



# Quantifying the effect of ocean bed properties on ice sheet geometry over 40,000 years with a full-Stokes model

Clemens Schannwell[1], Reinhard Drews[1], Todd A. Ehlers[1], Olaf Eisen[2,3], Christoph Mayer[4], Mika Malinen[5], Emma C. Smith[2], and Hannes Eisermann[2]

[1]Department of Geosciences, University of Tübingen, Tübingen, Germany
[2]Glaciology, Alfred Wegener Institute, Helmholtz Centre for Polar and Marine Research, Bremerhaven, Germany
[3]Department of Geosciences, University of Bremen, Bremen, Germany
[4]Bavarian Academy for Sciences and Humanities, Munich, Germany
[5]CSC-IT Center for Science Ltd., Espoo, Finland

**Correspondence:** Clemens Schannwell (Clemens.Schannwell@mpimet.mpg.de)

**Abstract.** Simulations of ice sheet evolution over glacial cycles requires integration of observational constraints using ensemble studies with fast ice sheet models. These include physical parameterisations with uncertainties, for example, relating to grounding line migration. Ice dynamically more complete models are slow and have thus far only be applied for <1,000 years, leaving many model parameters unconstrained. Here we apply a 3D thermomechanically coupled full-Stokes ice sheet model
to the Ekström Ice Shelf embayment, East Antarctica, over a full glacial cycle (40,000 years). We test the model response to differing ocean bed properties that provide an envelope of potential ocean substrates seawards of today's grounding line. The end member scenarios include a hard, high friction ocean bed and a soft, low friction ocean bed. We find that predicted ice volumes differ by >50% under almost equal forcing. Grounding line positions differ by up to 49 km, show significant hysteresis, and migrate non-steadily in both scenarios with long quiescent phases disrupted by leaps of rapid migration. The
simulations quantify evolution of two different ice sheet geometries (namely thick and slow vs. thin and fast), triggered by the variable grounding line migration over the differing ocean beds. Our study extends the timescales of 3D full-Stokes by an order of magnitude to previous studies with the help of parallelisation. The extended time frame for full-Stokes models is a first step towards better understanding other processes such as erosion and sediment redistribution in the ice shelf cavity impacting the entire catchment geometry.

## 1 Introduction

Shortcomings in the description of ice dynamics remain one of the limitations for projecting the evolution of the Greenland and Antarctic ice sheets (Pachauri et al., 2014). If current sea level rise rates continue unabated, up to 630 million people will be at annual flood risk by 2100 (Kulp and Strauss, 2019), making improved ice sheet model projections important to assess the socioeconomic impact. Due to the high computational costs of full-Stokes (FS) models that solve the complete ice dynam-
ical equations, current long term (>1,000 years) ice sheet simulations rely on simplifications to the ice dynamical equations. This choice is justified because it allows for ensemble modelling and tuning of unknown parameters using observations. There



are two drawbacks to this approach. First, it is uncertain whether the transition zone between grounded and floating ice is adequately represented in existing long term simulations (Pattyn and Durand, 2013). Second, the omission of membrane and bridging stress gradients hamper disentangling the relative contributions of basal sliding and ice deformation to the column

averaged ice discharge (MacGregor et al., 2016; Bons et al., 2018). The former is one of the main uncertainties for projecting the sea level contribution of contemporary ice sheets (Durand et al., 2009; Pattyn and Durand, 2013). The latter is a bottleneck for the inclusion of basal processes such as erosion and deposition of sediments which critically depend on the magnitude of basal sliding (e.g. Humphrey and Raymond, 1994; Egholm et al., 2011; Herman et al., 2011; Yanites and Ehlers, 2016; Alley et al., 2019) and may govern the formation and decay of ice streams (Spagnolo et al., 2016).

A number of simplified model variants of the full ice flow equations have been successfully applied to sea level rise reconstructions over timescales of >1,000 years (e.g. Golledge et al., 2012; Briggs et al., 2014; Pollard et al., 2016). In order to reproduce past ice sheet geometries paleo ice sheet models rely on observations that constrain the lateral as well as the vertical extent of the ice sheet (e.g. Briggs et al., 2014; Bentley et al., 2014; Golledge et al., 2014). Ice sheet extent is commonly inferred from marine sediment core data or geomorphological data, ice sheet elevation from exposure dating, and changes in ice thickness

from ice cores or ice rises (e.g. Bentley et al., 2010; Golledge et al., 2013; Briggs et al., 2014). Fast paleo ice sheet models employ ensemble simulations in which poorly known model parameters are tuned such that they match the constraints. This allows to gauge the uncertainties regarding for example atmospheric and oceanic boundary conditions over glacial cycle timescales (e.g. Golledge et al., 2012; Briggs et al., 2014; Pollard et al., 2016; Albrecht et al., 2020). Each ensemble member simulation is then evaluated against the constraints present at that particular timeslice. To determine the goodness of the fit of individual

ensemble members, modelling studies apply statistical methods ranging from weighted scoring schemes (e.g. Briggs et al., 2014; Albrecht et al., 2020) to statistical emulators (Pollard et al., 2016). The rationale behind this tuning is that if the model matches the constraints well, then confidence is high that the model also reproduces ice sheet changes at other times. The risk involved is that the matching may overcompensate for the simplified model physics leading to higher uncertainties in future predictions where model constraints are absent. Due to the high computational demands, both, in terms of mesh resolution and

the physics required to solve for a freely evolving grounding line (Gillet-Chaulet et al., 2012; Seddik et al., 2012; Favier et al., 2014; Schannwell et al., 2019), FS models up to now have been restricted to individual simulations and simulation lengths of <1,000 years for real world geometries. Therefore, there is a need to extend the applicability of regional FS ice sheet models to timescales longer than 1,000 years so that uncertainties due to physical approximations can be reduced.

For glacial cycle simulations with an advance and a retreat phase, the particular challenge arises that the ice sheet advances and

retreats over ocean beds where bathymetry and its geological properties are often poorly known. Ensemble modelling studies identified basal properties of ocean beds as a major source of uncertainty in ice dynamic models (e.g. Pollard and DeConto, 2009; Pollard et al., 2016; Whitehouse et al., 2017; Albrecht et al., 2020). This holds especially for drainage basins where such geological constraints are absent. Under contemporary ice sheets, estimating basal friction parameters (e.g. basal friction between the ice sheet and the underlying substrate) is virtually impossible by direct measurements and can only be inferred

indirectly on a continental scale by solving an optimisation problem matching today's surface velocities and/or ice thickness (e.g. MacAyeal, 1993; Gillet-Chaulet et al., 2012; Cornford et al., 2015). Furthermore, the inferred basal friction coefficient is





often spatially heterogeneous and can vary by up to five orders of magnitude under the present day Antarctic ice sheet (Cornford et al., 2015). To what extent this variability truly reflects variability in geology and/or hydrology, or is falsely introduced by the approximations in the ice dynamical equations or omission of ice anisotropy is unknown.

Here, we present the first regional scale FS simulations investigating the effect of different ocean bed properties on ice sheet geometry over a glacial cycle. We hereby extend the feasibility of regional FS ice sheet simulations by an order of magnitude using the open source code Elmer/Ice (Gagliardini et al., 2013). We do this with a highly parallelised numerical scheme allowing to maintain a high mesh resolution (∼1 km) and a freely evolving grounding line over glacial/interglacial timescales. Our simulations focus on the effect of ocean bed properties seawards of today's grounding line and to quantify their impact on the

evolution of the entire catchment. This is done for the Ekström Ice Shelf catchment, Dronning Maud Land, East Antarctica (Fig. 1).

## 2   The Ekström catchment, Dronning Maud Land, East Antarctica

We have chosen the Ekström catchment for our study because it hosts the German overwinter station Neumayer III and is therefore particularly well constrained by geophysical and climatological observations and boundary datasets. Uncertainties in

the contemporary ice sheet geometry are small because of previous dense airborne radar surveys (Fretwell et al., 2013). Unlike many other ice shelves, the bathymetry in this area is known to an unprecendented extent from seismic reflection surveying (Smith et al., 2019). This has been complemented with bathymetry modelling via gravity inversion from airborne gravity data to cover the whole cavity (Eisermann et al., 2020, in review). In comparison to the Bedmap2 dataset (Fretwell et al., 2013), the updated cavity is up to 1,000 m deeper. We use the Eastern Dronning Maud Land (EDML) ice core (Graf et al., 2002)

as proxy for past temperature variations in the region. The location of the EDML ice core is about 700 km to the south-east of the modelling domain on the Antarctic plateau. The Ekström catchment contains also two ice rises (Schannwell et al., 2019; Drews et al., 2013) with independent ice flow centres from the main ice sheet. Ice rises archive the regional ice sheet history in their internal stratigraphy. Therefore, their stability or lack thereof provides indications about past ice flow changes of the area. Furthermore, while geological constraints about the retreat history since the LGM are still uncertain, there is

evidence in this area from multiple geophysical observations (Kristoffersen et al., 2014) and geological signatures (Eisermann et al., 2020, in review) about contrasting ocean bed properties. There is also growing evidence that the catchment is close to steady state (e.g. Drews et al., 2013; Schannwell et al., 2019) which we consider beneficial for our model initialisation. While much recent research has focused on the fast flowing outlet glaciers of Antarctica, we stress the importance of also studying catchments characterised by slower moving ice (<300 m/yr), as they occupy ∼90% of the contemporary Antarctic grounding

line and account for 30% of the total ice discharge (Bindschadler et al., 2011; Rignot et al., 2011). The results we obtain for the Ekström Ice Shelf catchment could therefore be relevant for many other catchments around Antarctica and hence the total budget.



**Figure 1.** Overview of the Ekström Ice Shelf catchment with present day grounding line (Bindschadler et al., 2011) and model domain. Cyan square shows location of Neumayer Station III. Filled black circles indicate location of ice rises. Flowline (A-A') is shown in Fig. 10. Background is the MODIS Mosaic of Antarctica (Scambos et al., 2007).

## 3 Model description

### 3.1 Ice flow equations

Ice flow is dominated by viscous forces which permits the dropping of the inertia and acceleration terms in the linear momentum equations. The Elmer/Ice ice sheet model (Gagliardini et al., 2013) solves the complete 3D equation for ice deformation. This results in the Stokes equations described by

$$\nabla \cdot \boldsymbol{\sigma} = -\rho_i \boldsymbol{g}. \tag{1}$$

Here, $\boldsymbol{\sigma} = \boldsymbol{\tau} - p\boldsymbol{I}$ is the Cauchy stress tensor, $\boldsymbol{\tau}$ is the deviatoric stress tensor, $p = -tr(\boldsymbol{\sigma})/3$ is the isotropic pressure, $\boldsymbol{I}$ the

identity tensor, $\rho_i$ the ice density, and $\boldsymbol{g}$ is the gravitational vector. Ice flow is assumed to be incompressible which simplifies



mass conservation to

$$\nabla \cdot \boldsymbol{u} = 0, \tag{2}$$

with $\boldsymbol{u}$ being the ice velocity vector. Here we model ice as an isotropic material. Its rheology is given by Glen's flow law which relates the deviatoric stress tensor $\boldsymbol{\tau}$ with the strain rate tensor $\dot{\boldsymbol{\epsilon}}$:

$$\boldsymbol{\tau} = 2\eta\dot{\boldsymbol{\epsilon}}, \tag{3}$$

where the effective viscosity $\eta$ can be expressed as

$$\eta = \frac{1}{2}B\dot{\epsilon}_e^{\frac{(1-n)}{n}}. \tag{4}$$

In this equation $B$ is a viscosity parameter that depends on ice temperature relative to the pressure melting point computed through an Arrhenius law, $n$ is Glen's flow law parameter ($n$=3), and the effective strain rate is defined as $\dot{\epsilon}_e^2 = tr(\dot{\boldsymbol{\epsilon}}^2)/2$.

## 3.2 Ice temperature

The ice temperature is determined through the heat transfer equation (e.g. Gagliardini et al., 2013) which reads

$$\rho_i c_v \left( \frac{\partial T}{\partial t} + \boldsymbol{u} \cdot \nabla T \right) = \nabla \cdot (\kappa \nabla T) + \dot{\boldsymbol{\epsilon}} : \boldsymbol{\sigma}, \tag{5}$$

where $c_v$ and $\kappa$ are the specific heat of ice and the heat conductivity, respectively. The **:** operator represents the colon product between two tensors. This last term of the equation represents strain heating.

## 3.3 Boundary conditions

### 3.3.1 Ice temperature

Our parameterisation of surface temperature changes follows Ritz et al. (2001). We parameterise relative surface temperature changes to present day as a function of relative surface elevation change with respect to present day elevations and a spatially uniform surface temperature variation that is derived from the nearby EDML ice core (Graf et al., 2002). The surface

temperature is then given by (Ritz et al., 2001, eq. 11):

$$T_a = T_{a0} - \gamma_a(z_{s0} - z_s) + \Delta T_{clim}. \tag{6}$$

Here, $T_a$ and $T_{a0}$ are the surface temperatures at the current timestep and present day. The present day temperature distribution is taken from Comiso (2000). $z_s$ and $z_{s0}$ are the surface elevations at the current timestep and present day, and $\Delta T_{clim}$ is the climatic forcing derived from the EDML ice core. As in Ritz et al. (2001), we apply a spatially constant lapse rate ($\gamma_a$) of

0.00914 K/m (Table 1).

At the grounded base of the ice sheet, where the ice is contact with the subglacial topography, we prescribe the geothermal heat flux (Martos et al., 2017). This heat flux is time invariant. Ice temperature is set to the local pressure melting point for the boundary condition underneath the floating ice shelves.



### 3.3.2 Surface mass balance (SMB) and basal mass balance (BMB)

A kinematic boundary condition determines the evolution of upper and lower surfaces $z_j$:

$$\frac{\partial z_j}{\partial t} + u_x \frac{\partial z_j}{\partial x} + u_y \frac{\partial z_j}{\partial y} = u_z + \dot{a}_j, \tag{7}$$

where $\dot{a}_j$ is the accumulation/ablation term and $j = (b, s)$, with $s$ being the upper surface and $b$ being the lower surface (base) of the ice sheet.

For the surface mass balance (SMB) parameterisation, we closely follow Ritz et al. (2001) again. We assume that no melt

occurs in all our simulations. This is justified because SMB models simulate little melt at present day conditions (Lenaerts et al., 2014) and these are the warmest years in our simulations. As for the surface temperature, our SMB parameterisation uses a present day distribution of the SMB (Lenaerts et al., 2014) as input. Variations of the SMB over time are then proportional to the exponential of the surface temperature variation (Ritz et al. (2001), eq. 12):

$$\dot{a}_s(T_a) = a_{s0}(T_{a0}) exp(\Delta a (T_a - T_{a0})), \tag{8}$$

where $a_{s0}$ is the present day SMB, $a_s$ is the SMB at the current timestep, and the parameter $\Delta a = 0.07 \text{ K}^{-1}$. This means that for a surface temperature drop of 10 K, the SMB is reduced by 50% (Ritz et al., 2001).

Sub shelf melting underneath the floating ice shelves is based on the difference between the local freezing point of water under the ice shelves and the ocean temperature near the continental shelf break (Beckmann and Goosse, 2003). The freezing temperature ($T_f$) is calculated through:

$$T_f = 0.0939 - 0.057 S_o + 7.64 \times 10^{-4} z_b, \tag{9}$$

where $z_b$ is the base of the ice shelf and $S_o$ is the ocean salinity (Table 1). The basal melt rates ($\dot{a}_b$) are then computed by

$$\dot{a}_b = \frac{\rho_w c_{p_o} \gamma_T F_{melt} (T_O - T_f)^2}{L \rho_i}. \tag{10}$$

In this equation, $\rho_w$ is the density of water, $c_{p_o}$ is the specific capacity of the ocean mixed layer, $\gamma_T$ is the thermal exchange velocity, $L$ is the latent heat capacity of ice, $F_{melt}$ is a tuning parameter to match present day melt rates, and $T_O$ is the ocean

temperature (Table 1). The ocean temperature is initially set to $-0.52°\text{C}$ (Beckmann and Goosse, 2003). $F_{melt}$ is chosen such that present day basal melt rates do not exceed $\sim$1.1 m/yr. This is in accordance with melt rates derived from satellite observations and mass conservation (Neckel et al., 2012). Applied variations of the ocean temperature are a damped ($\sim$40%) and delayed ($\sim$3,000 years) version of the climatic forcing for surface temperature $\Delta T_{clim}$ (Bintanja et al., 2005).

### 3.3.3 Basal sliding and sea level

Where the ice is in contact with the subglacial topography a linear Weertman-type sliding law of the form

$$\boldsymbol{\tau_b} = C |\boldsymbol{u_b}|^{m-1} \boldsymbol{u_b}, \tag{11}$$





is employed. Here $\tau_b$ is the basal traction, $m$ is the basal friction exponent which is set to 1 in all simulations, and $C$ is the basal friction coefficient. A linear viscous sliding relation ($m$=1) was chosen to guarantee consistency between model intialisation and forcing simulation. The consequences of this choice on the results are discussed below (see section 5.5). For the present

day grounded ice sheet, $C$ is inferred by solving an inverse problem (see section 3.4), and for the present day ocean beds a uniform basal friction coefficient of $10^{-1}$ MPa m$^{-1}$ yr and $10^{-5}$ MPa m$^{-1}$ yr is prescribed for the soft (sediment based) bed and hard (crystalline rock based) bed simulations. Underneath the floating part of the domain basal traction is zero ($\tau_b = 0$), but hydrostatic sea pressure is prescribed. We initialise the present day sea level to zero and apply sea level variations according to Lambeck et al. (2014).

**Table 1.** Numerical values of the parameters adopted for the simulations

| Parameter | Symbol | Value | Unit |
|---|---|---|---|
| Ice density | $\rho_i$ | 917 | kg m$^{-3}$ |
| Ocean density | $\rho_w$ | 1028 | kg m$^{-3}$ |
| Glen's exponent | n | 3 | |
| Gravity | g | 9.81 | m s$^{-2}$ |
| Atmospheric lapse rate | $\gamma$ | 0.00914 | K m$^{-1}$ |
| Tuning parameter SMB | $\Delta$a | 0.07 | K$^{-1}$ |
| Ocean salinity | $S_0$ | 35.0 | PSU |
| Heat capacity | $c_{p_o}$ | 3974 | J kg$^{-1}$ °C$^{-1}$ |
| Latent heat of fusion | L | $3.35 \times 10^{-4}$ | J kg$^{-1}$ |
| Tuning parameter BMB | $F_{melt}$ | $0.383 \times 10^{-4}$ | |
| Thermal exchange velocity | $\gamma_T$ | $1 \times 10^{-5}$ | m s$^{-1}$ |

### 3.4   Model initialisation

The model is initialised to the present day geometry using the commonly applied snapshot initialisation in which the basal traction coefficient $C$ is inferred by matching observed surface velocities with modelled surface velocities. We take advantage of the quasi steady state of the catchment and use same optimisation parameters as in Schannwell et al. (2019). Similar to Zhao et al. (2018), we employ a two step initialisation scheme. In the first iteration, the optimisation problem is solved with

an isothermal ice sheet with ice temperature set to –10°C. The resulting velocity field is then used to solve the steady state temperature equation before the optimisation problem is solved again with the new temperature field. This type of temperature initialisation approach provides similar results to a computationally more expensive temperature spin up over several glacial cycles (Rückamp et al., 2018), as long as the system is close to steady state.



## 3.5 Mesh generation and refinement

We initially create a 2D isotropic mesh with a nominal mesh resolution of ∼6 km everywhere in the domain. To ensure that we
simulate grounding line dynamics at the required detail, we use the meshing software MMG (http://www.mmgtools.org/, last
access: 28 February 2020) to locally refine the mesh down to ∼1 km in the region of present day Ekstöm Ice Shelf (Figure 2)
with areas away from the region of interest remaining at ∼6 km resolution. The mesh is then vertically extruded, consisting of
10 layers and the horizontal mesh size is kept constant throughout the simulations.

## 175 3.6 Block preconditioned ParStokes solver

Because of the non-Newtonian rheology of ice and the dependence of viscosity on strain rates, the resulting Stokes equations
are non-linear and have to be solved iteratively. In three dimensions the arising systems of linear equations become large ($10^6$–
$10^7$ degrees of freedom) at high mesh resolution. Standard iterative methods (Krylov subspace methods) in conjunction with
algebraic preconditioners (e.g. Incomplete Lower Upper (ILU) decomposition) do often not converge for real world geometries
in glaciology. High aspect ratios of the finite elements and spatial viscosity variations of several orders of magnitudes, strongly
affect accuracy and stability of the numerical solution (Malinen et al., 2013). This means that most glaciology applications with
Elmer/Ice revert to using a direct method for solving the Stokes equations. While robust, direct solvers do not take advantage
of the sparse structure of the matrix and require large amounts of memory. In three dimensions their memory requirements
increase with the square of the number of unknowns. Therefore, we use a stable parallel iterative solver (ParStokes) in our
simulations that is implemented in Elmer/Ice, but has so far been rarely used. ParStokes is based on block preconditioning
(Malinen et al., 2013) that improves the solvability of the underlying saddle point problem through clustering of Eigenvalues.
As we will show below the Krylov subspace methods now converge better and lead to improved scaling with more Computer
Processing Units (CPUs).




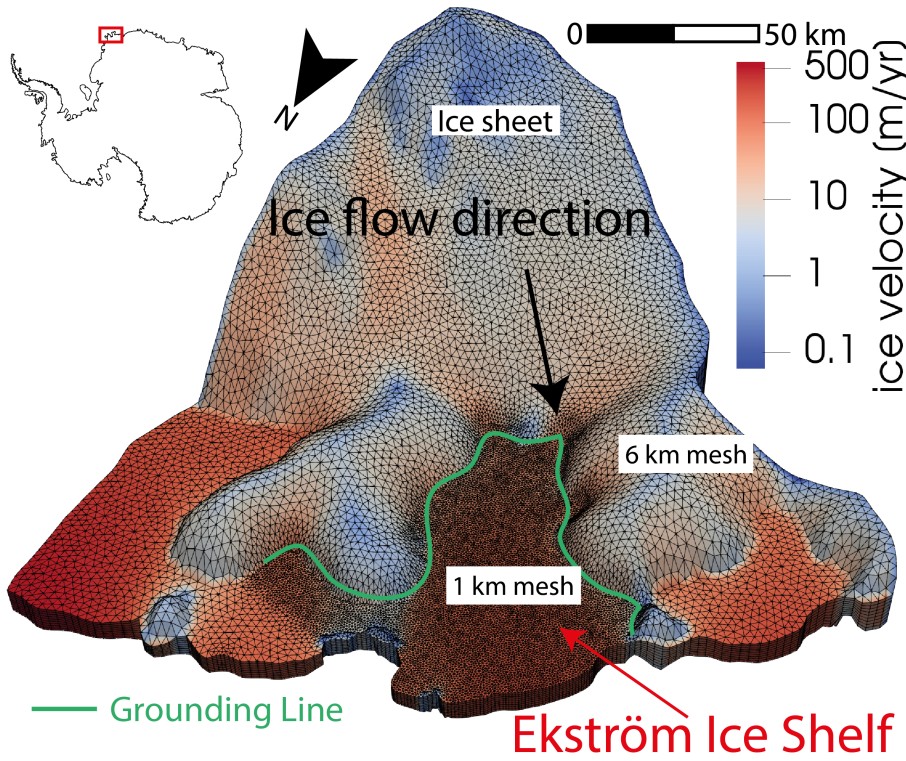

**Figure 2.** Model domain of Elmer/Ice in 3D including numerical mesh of Ekström Ice Shelf catchment, East Antarctica, with ice velocity in the background

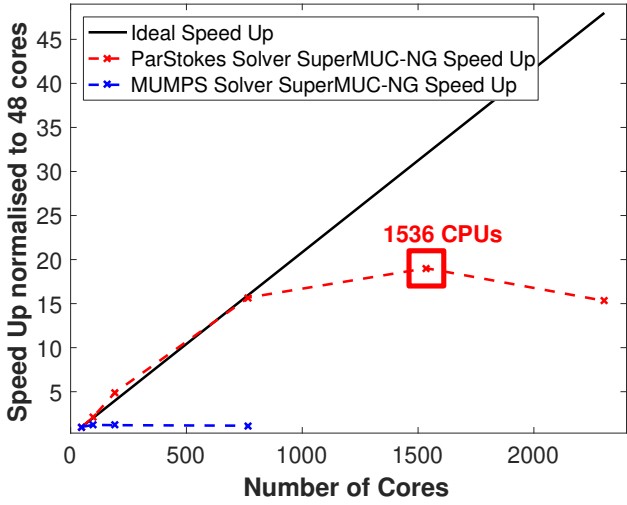

**Figure 3.** Scaling behaviour of iterative solver (ParStokes) and direct Solver (MUMPS) for Elmer/Ice on the SuperMUC-NG supercomputer. Red square denotes number of CPUs selected for this study.





### 3.7   Experimental design

We demonstrate a FS simulation of ice sheet growth and decay over 40,000 years. During the first 20,000 years the atmospheric and oceanic forcing simulates the transition from an interglacial to a glacial (henceforth called the advance phase). We then symmetrically reverse the climate forcing to simulate deglaciation (henceforth called the retreat phase). The symmetrical reversal of the model forcing enables investigation of hysteresis effects. The interglacial starting conditions are chosen with present day properties and characteristics, so that the best possible basal friction coefficient beneath the grounded ice sheet can

be found using today's ice sheet geometry and surface velocities (Schannwell et al., 2019). The glacial conditions are chosen to resemble the Last Glacial Maximum for which we have good constraints for atmospheric forcing from the nearby EDML ice core. We consider two end member basal property scenarios by prescribing either soft ocean bed conditions (mimicking sediment deposits) or hard ocean bed conditions (mimicking crystalline rock) for all present day ocean cavities in the modelling domain. The tested scenarios of basal traction coefficients encompasses what other ice sheet models have inferred (e.g.

Cornford et al., 2015) for the grounded portion underneath the present day Antarctic ice sheet (basal traction coefficient ranging from $10^{-1}$ MPa m$^{-1}$ yr for sediments to $10^{-5}$ MPa m$^{-1}$ yr for crystalline bedrock). Those end member values do not reflect a true range of sliding coefficients for a given sliding law, but were derived as tuning parameters. Hence they also account for uncertainties in model parameters, forcings, and physics of the applied ice sheet model. That is why we consider those values as end members and regard simulated differences in ice volume and grounding line position as the maximum envelope

of uncertainties resulting from different ocean bed properties. We perform the simulations with a) the standard Elmer/Ice setup using the Multifrontal Massively Parallel Sparse (MUMPS) direct solver for ice velocities; and b) using a stable iterative solver for ice velocities (see section 3.6), resulting in a total of four simulations. We carried out the simulation on three different high performance computing systems: the ZDV cluster, the now decommissioned SuperMUC system, and the SuperMUC-NG system.

## 4   Results

The results can be divided into methodological advances and new scientific insights. In the following, we first present the technical improvements of the presented Elmer/Ice model setup in comparison to the "classic" setup employed in previous studies (e.g. Schannwell et al., 2019). This is followed by the analysis of the performed model simulations in terms of ice flow behaviour and an analysis of the role of the subglacial strata characteristics for advance and retreat dynamics.

### 4.1   Comparison between direct Stokes solver (MUMPS) and ParStokes

The ParStokes solver allows for a much better scaling of the required computation time with increasing numbers of CPUs (Figure 3). While there is no speed up for the "classic" solver setup using the direct solver MUMPS, there is a linear speedup for the ParStokes solver up to ∼700 CPUs before the rate of speedup tapers off and vanishes for more than 1536 CPUs. This much better scaling behaviour results in a total compute time for the iterative solver on the SuperMUC-NG system that is



faster between a factor 3–6 in comparison to the MUMPS solver setup on the ZDV system. For our simulations, this means
that the 40,000 year simulation now takes 23 days instead of 141 days for the hard bed case, and 27 days instead of 94 days
for the soft bed case (Figure 4). In comparison, on the now decommissioned SuperMUC system, total compute time savings
were only 20 days in comparison to the MUMPS solver setup. The reason for this were the long queuing intervals in between
simulations, leading to an additional >80 days of waiting for simulations to run in comparison to the other systems. This is a
direct consequence of the system being in the process of shutting down and hence only running at 50% capacity.

We use predicted grounding line position and ice thickness as metrics to compare the "classic" solver setup using MUMPS with
the new solver ParStokes. We note however that we do not expect a perfect match between the two solver setups due to small
differences in the finite element formulation (e.g. stabilisation method). For both simulations, there is good agreement in terms
of grounding line position over time, with differences never exceeding 5% (Figure 5). Because the soft bed simulation exhibits
smaller magnitude grounding line motion over the simulation, agreement between the two solver setups is better, with differ-
ences well below 1% for almost the entire simulation length. In the hard bed simulation, where larger magnitudes of grounding
line motion are predicted, the ParStokes solver's grounding line is not as far advanced as the MUMPS solver grounding line
(Figure 6). Moreover, at times of rapid grounding line motion, the response of the grounding line in the ParStokes solver is
delayed by up to ∼3,500 years. This leads to differences in transient grounding line positions (<5%). However grounding line
positions for steady state situation differ negligibly (<1.5% difference). The predicted ice thickness differences are larger, par-
ticularly for the hard bed run, where ice thickness change is larger overall. Locally these differences can be as large as ∼460 m
(<25% of the ice thickness) in transient scenarios. They are most pronounced in periods of delayed grounding line response.
Once a stable grounding line position has been reached, thickness differences are notably smaller (Figure 6, 7). Overall, the
ParStokes solver provides comparable results to the MUMPS solver, but is much superior in terms of the required computation
time. Therefore, the remainder of the results section will be based on the ParStokes solver simulations.




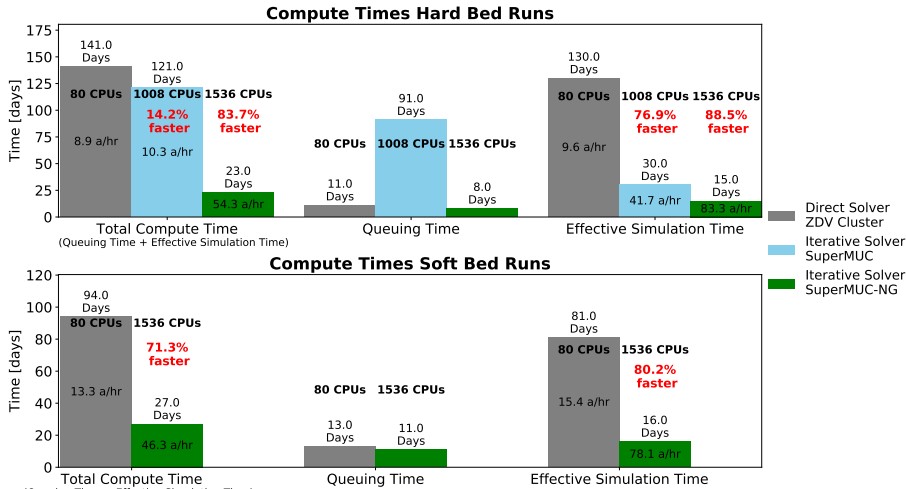

**Figure 4.** Speed up of iterative Solver (ParStokes, green and blue bars) in comparison to direct solver (MUMPS, gray bars) for the hard bed cavity (upper panel) and soft bed cavity (lower panel) simulations. Simulations were performed on three different high performance computing systems (ZDV, SuperMUC, and SuperMUC-NG).

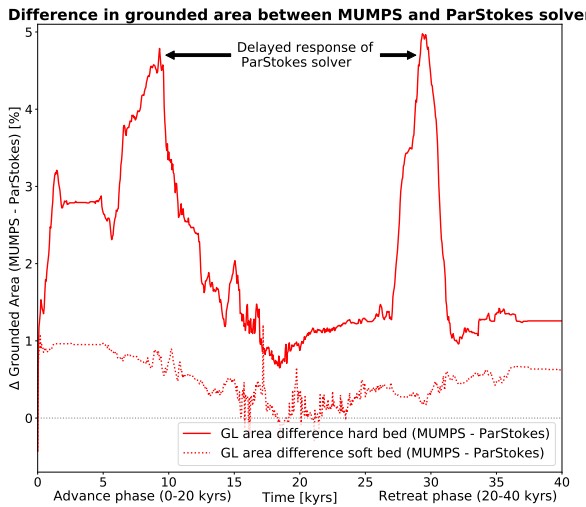

**Figure 5.** Differences in grounded area between the classic MUMPS and ParStokes solver setup for the soft bed and hard bed simulations.





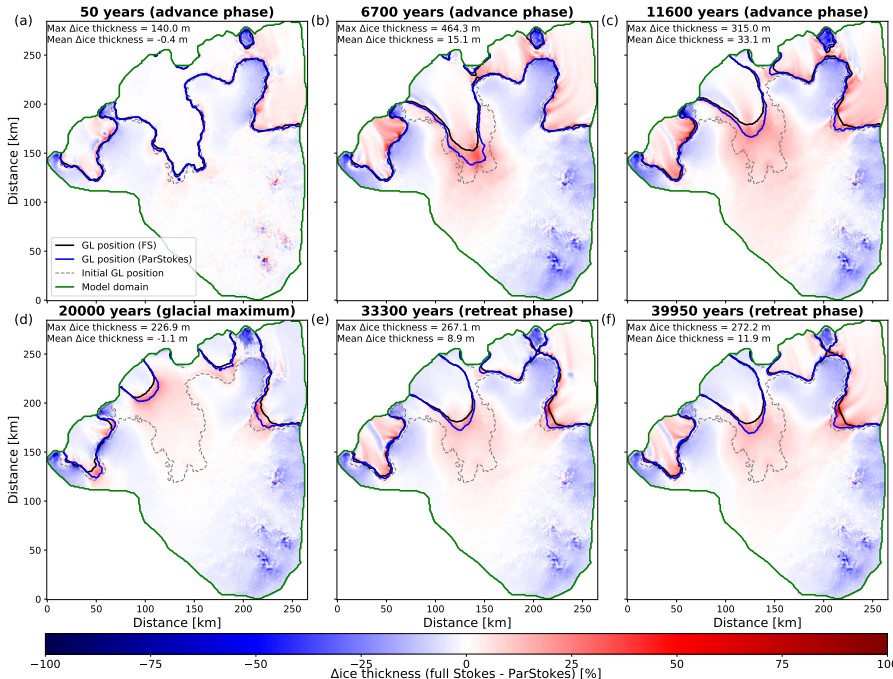

**Figure 6.** Differences in grounding line position and ice thickness between the classic MUMPS and ParStokes solver setup for the hard bed simulation at specific time slices.





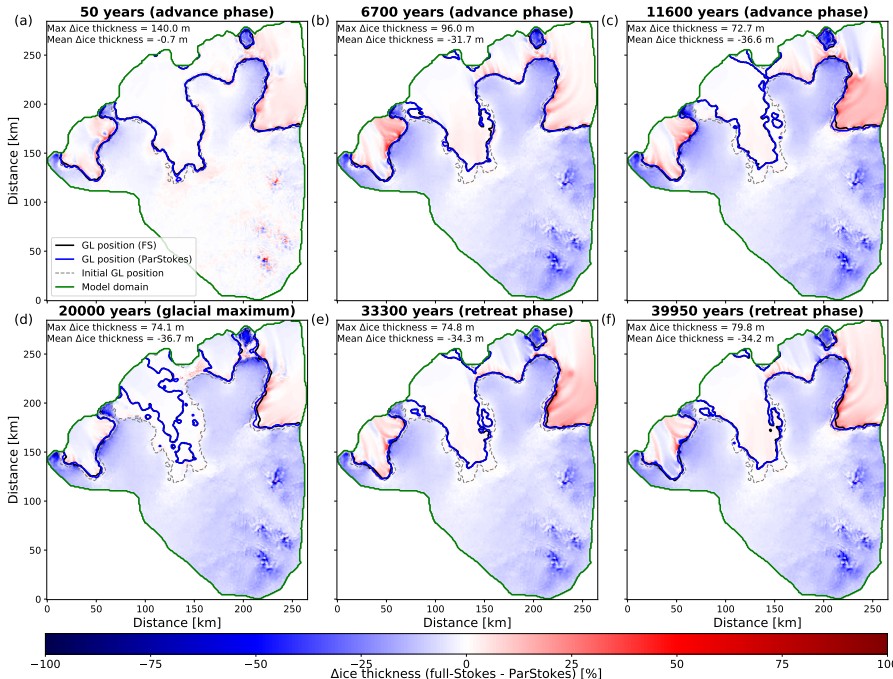

**Figure 7.** Differences in grounding line position and ice thickness between the classic MUMPS and ParStokes solver setup for the soft bed simulation at specific time slices.





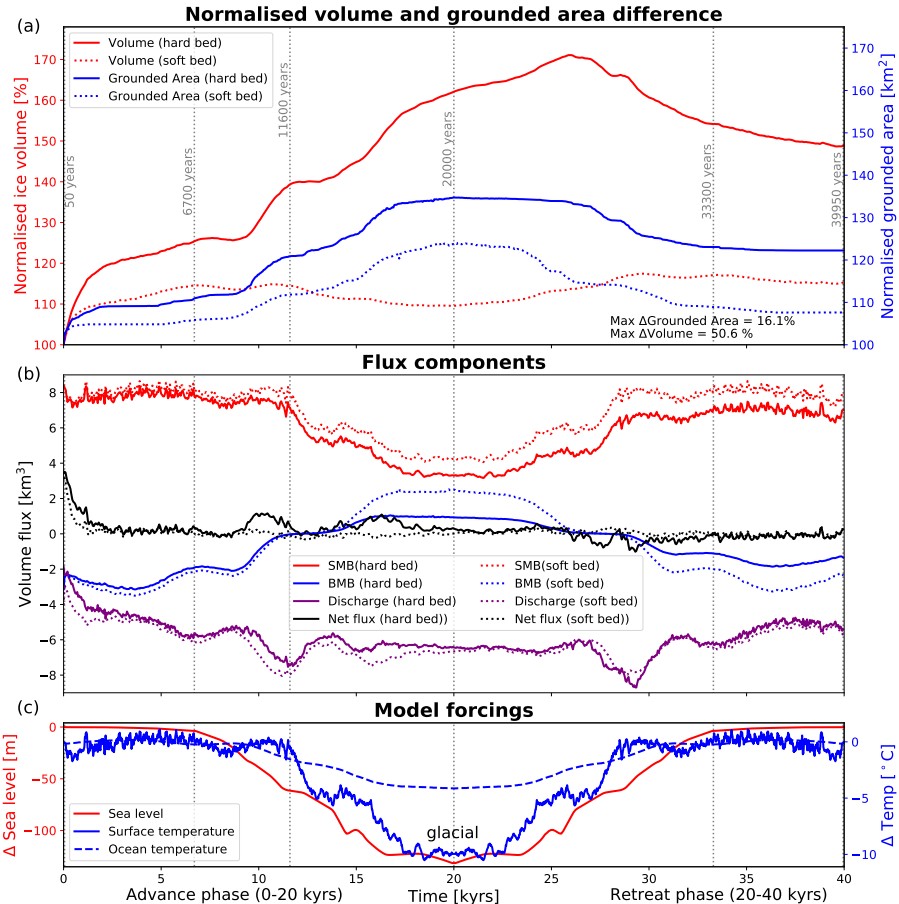

**Figure 8.** Ice sheet evolution and model forcing for soft and hard bed simulations. (a) shows volume and grounded area evolution normalsied to present day. (b) shows corresponding mass balance fluxes, and (c) shows most important model forcings. Vertical grey stippled lines show time slices shown in Figs. 6, 7, 9, and 10.



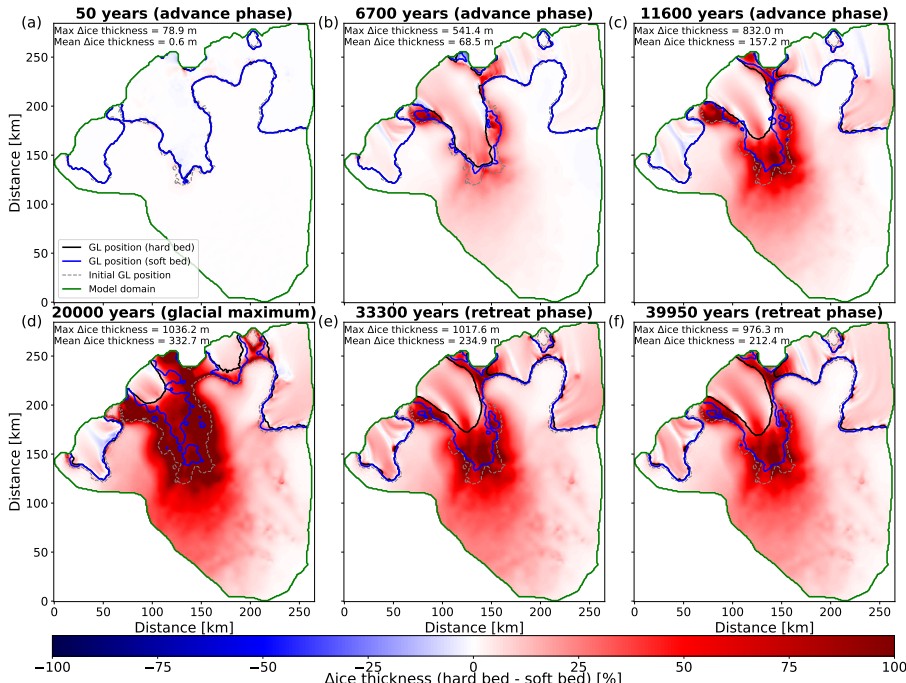

**Figure 9.** Differences in plane view of ice thickness and grounding line positions between the hard and soft bed simulations at selected time slices. (a-d) show differences in the advance phase and (e,f) show differences in the retreat phase.

## 4.2 Influence of bed hardness on ice sheet growth and decay

As expected, the hard and soft bed simulations result in different ice sheet geometries. Quantitatively, both scenarios differ significantly in transient and steady state volumes (Fig. 8), fluxes, and grounding line positions (Figs. 9 and 10). The simulated hard bed ice sheet is in many areas more than twice as thick as the soft bed ice sheet, with maximum ice thickness differences

between hard and soft bed reaching 1,036 m or 120% (Fig. 10). In more detail, the differences between these simulations are as follows. First, the hard bed ice sheet results in a thick, slow, and large volume ice sheet after 20,000 years at glacial conditions. During the advance phase, volume increases occur step-wise with three distinct periods of volume increases (Fig. 8). These periods of volume increase in the region of interest are short (<2,000 years) and are interrupted by longer periods of little ice volume change. At the glacial maximum, the volume increase in comparison to the interglacial is ∼60%. During the first

∼8,000 years in the retreat phase, the hard bed simulation continues to gain volume albeit at a slow rate. In the following the ice sheet starts to loose volume. However, the rate of volume loss is small, such that after a full glacial cycle, the total ice volume is still ∼47% more of what is was at the beginning of the simulation.

Second, unlike the hard bed simulations, the soft bed simulation leads to a thin, fast, and small volume ice sheet at glacial conditions. During the advance phase, this simulation does not show a step-wise volume gain pattern. In fact, apart from an

initial volume gain in the first 1,000 years of the advance phase (∼10%), there is very little volume change. This leads to a volume increase of merely ∼8% at the glacial maximum. The trend of little volume variations continues during the retreat





phase, where in the first 10,000 years a volume increase of ∼8% occurs, before the volume remains approximately constant for the remainder of the retreat phase.

The entirely different ice sheet geometries for soft and hard bed simulations have consequences for the two ice rises present in
the catchment (Fig.1). While both ice rises and their divide positions are very little affected by the soft bed simulations, they are partly overrun in the hard bed simulation such that their local ice flow centre vanishes (SI video 1).

## 4.3   Grounding line and ice sheet stability

Stable grounding line positions for both simulations are associated with periods of ice sheet stability (Fig. 8). There are three
distinct periods of grounding line stability in the advance phase and one period of grounding line stability in the retreat phase. All of these four periods are longer than 3,000 years. Periods of grounding line advance in comparison are characterized by short leaps taking no longer than 1,000–2,000 years (Fig. 8). During the advance phase, differences in grounding line positions between the hard bed and soft bed simulations gradually increase from 7 km after ∼1,500 years to over 37 km after 11,600 years, and finally to its maximum difference of 49 km at the glacial maximum (Fig. 10). Grounding line advance for the hard
bed is more than twice as far (∼110% larger) than its soft bed counterpart in the advance phase. In the retreat phase, the soft bed simulation shows higher grounding line fidelity compared to the hard bed simulation. The soft bed starts to exhibit grounding line retreat after ∼4,000 years into the retreat phase, whereas the hard bed does not show grounding line retreat for ∼8,000 years into the retreat phase.

## 4.4   Hysteresis of ice sheet simulations

Next we compare the ice sheet geometries during a full glacial cycle in which atmospheric and oceanic forcing are essentially symmetrically reversed. There is a significant grounding line advance in the first ∼300 years in both simulations. In the following, hysteresis is analysed with respect to this position, rather than the start of the simulation. Only the hard bed simulation shows significant hysteresis behaviour, while the soft bed simulation has negligible hysteresis (Fig. 11). For the hard bed simulation, the grounding line after a full glacial cycle is ∼38 km further downstream of its initial position. This means that during
the retreat phase, the grounding line retreats only ∼48% in comparison to the simulated grounding line advance during the retreat phase of the hard bed simulation.





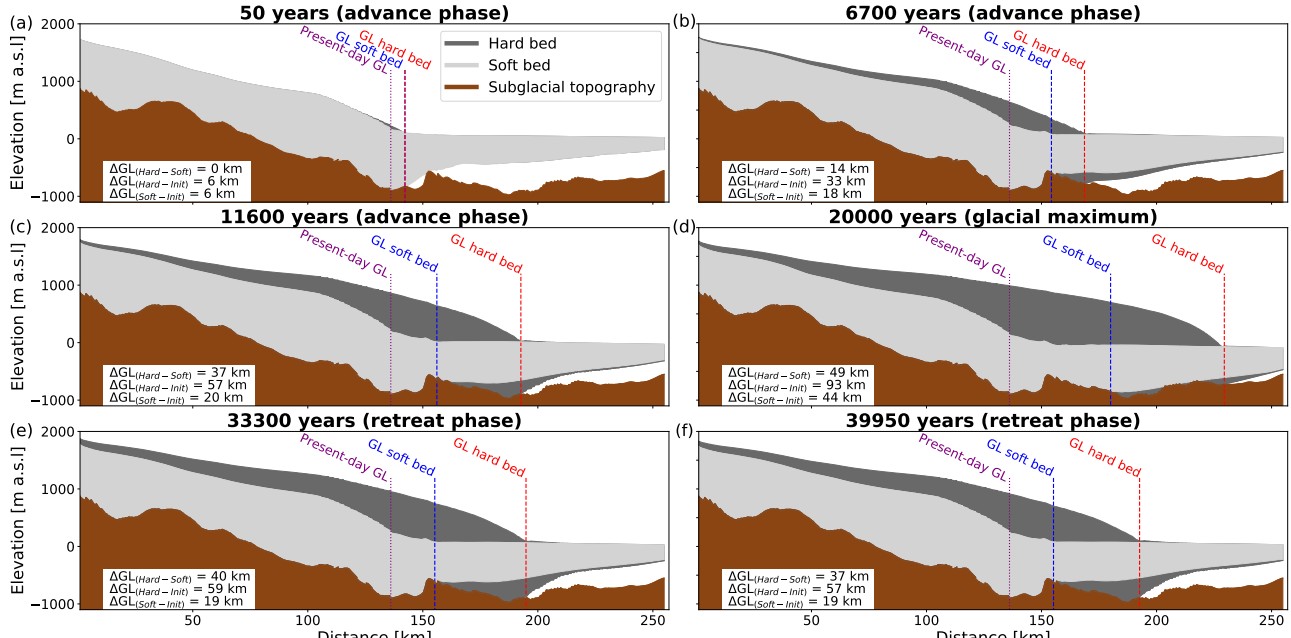

**Figure 10.** Difference in ice sheet geometry and grounding line position along a flowline (A-A' in Fig. 1) for the soft and hard bed simulations. (a-d) show differences in the advance phase and (e,f) show differences in the retreat phase.

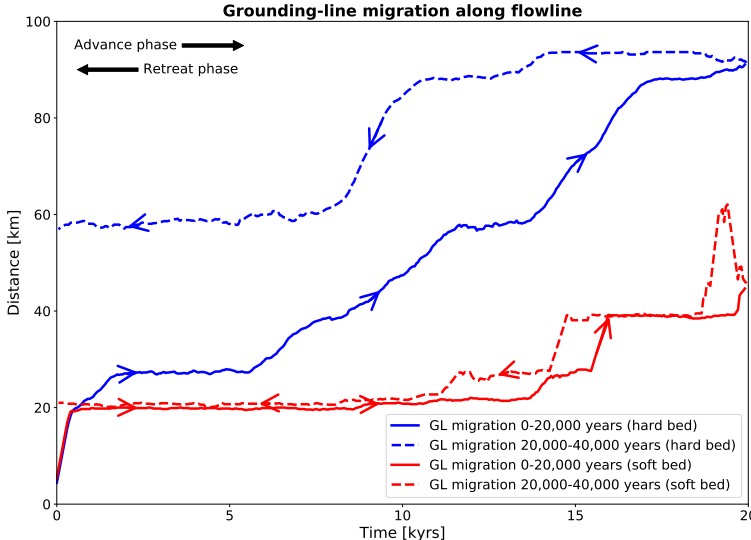

**Figure 11.** Grounding line migration along a flowline (A-A' in Fig. 1) for the soft and hard bed simulations for the advanced (solid lines) and retreat phase (dashed lines).





## 5 Discussion

### 5.1 Extending the feasibility timescales of full-Stokes models

The inclusion of the iterative ParStokes solver results in a speed up by a factor 3-6 compared to the direct solver. While ground-
ing line positions agree well between the two solver setups, during periods of rapid grounding line migration, positions can
differ by up to ∼5%. We note, however, that we do not expect a perfect match between the two solver setups due to small dif-
ferences in the finite element formulation (e.g. stabilisation method). Therefore, differences in grounding line positions were
expected between the solver setup, but they turn out to be small. The new setup extends the time range of 3D full-Stokes ice
sheet models on the regional scale from ≤1,000 years previously to 40,000 years now. The high mesh resolution required
to adequately capture grounding line migration (Pattyn et al., 2013) is hereby maintained. However, while the time range is
now significantly extended, our modelling approach only brackets the effect of ocean bed properties. As detailed below (sec-
tion 5.5), many other factors influencing ice sheet evolution, such as the applied BMB and SMB parameterisations, and basal
sliding relation, remain poorly constrained or are even excluded (e.g. glacial isostatic adjustment). Ensemble modelling (e.g.
Golledge et al., 2012; Briggs et al., 2014; Golledge et al., 2014; Pollard et al., 2016; Albrecht et al., 2020) using simplified ice
physics is better suited for this, because these models can more easily include other important model sub systems (e.g. basal
hydrology, basal sliding) and evaluate their respective uncertainties.

Our efforts aim towards including higher order ice physics into paleo ice sheet simulations. The advantages of our FS simula-
tions are as follows. By retaining all terms in the force balance, we have a solid physical representation of internal deformation
and grounding line dynamics over glacial timescales. This permits an improved quantification of the relative contributions
from basal sliding and ice deformation to the column averaged ice discharge, opening the door for a better understanding of
basal processes such as erosion and deposition of sediments and the formation of ice streams. We are also able to quantify the
effect of ocean bed properties onto the grounded ice sheet as the backstress provided by the contrasting ocean bed properties is
correctly transmitted upstream by our FS model. Grounding line migration also needs to be interpreted in relation to observed
bedforms. For example, the bedrock bump at 150 km in Figure 10 is interpreted as a potential overdeepening, carved out by the
confluence of two paleo ice streams (Smith et al., 2019). Our study presents the numerical framework to test hypotheses such
as this. Even though we are still not able to constrain our model with paleo observations due to the computation requirements,
our study provides an important first step towards it. In addition, computing the full 3D ice velocity field from the linear mo-
mentum equations may help to include thus far unused paleo data as constrains. For example, radar isochrones for floating ice
shelves could be incorporated more easily into the model tuning, because the FS approach does not apply a vertical average
in these areas. Ensemble modelling and our approach are in that regard complimentary. Both approaches should be pursued as
improvements to either are mutually beneficial for both. This also holds for shallow ice approximation-FS hybrid approaches
(Ahlkrona et al., 2016) which can build on the results shown here.



## 5.2 Influence of bed hardness on ice sheet growth and decay

The completely different ice sheet geometries for the hard and soft bed simulations are a consequence of the different levels
of basal friction provided by the hard and soft bed, respectively. The predicted differences between the hard bed and soft bed
simulations underline the high significance of a proper choice of basal properties used for ocean beds. The higher basal friction
in the hard bed case leads to elevated back stress and corresponding dynamical thickening of the inland ice sheet far upstream
of the grounding line. Although the SMB and BMB forcings equally depend on the ice sheet geometry through the applied
parameterisations, these effects are small compared to the ice dynamically induced thickening (Fig. 9). This clearly shows that
in the absence of other forcing mechanisms, ocean bed properties exert an important control on ice sheet growth and decay.
The importance of ocean bed properties on ice sheet evolution is long known (e.g. Pollard and DeConto, 2009; Whitehouse
et al., 2012; Pollard et al., 2016; Whitehouse et al., 2017; Albrecht et al., 2020). Here we quantify upper and lower bounds
of this effect for the first time on a regional scale with a FS model. Our results indicate that spatial changes of basal friction
coefficients in the cavities are likely very important for ice sheet growth and decay behaviour. This is relevant for the Ekström
Ice Shelf embayment and probably most of Dronning Maud Land, as evidence from geophysical data show that the ocean
bed of the Ekström cavity consists at least partly of crystalline bedrock (Kristoffersen et al., 2014; Smith et al., 2019). This
feature is more than 1000 km long. A new compilation and interpretation of airborne geophysics data by Eisermann et al.
(2020, in review) shows that the northern edge of a strong magnetic anomaly coincides with the location of the outcrop of
the Explora Volcanic Wedge (Smith et al., 2019), where subglacial material changes from ocean sediments to crystalline rock.
This transition cross-cuts the Ekström Ice Shelf cavity from ENE to WSW over its full width. Based on our simulations, such
crystalline outcrops under ice shelves will result in a thicker but slower ice sheet over the last glacial cycle, compared to a thin
and fast ice sheet linked to soft ocean beds which are mostly assumed for areas that lie below present day sea level (Pollard
and DeConto, 2009; Pollard et al., 2016; Whitehouse et al., 2017). Interestingly, today's north-eastern most grounding line of
Halfvarryggen ice rise coincides with this magnetic anomaly and the Explora Volcanic Wedge outcrop and thus likely with the
presence of subglacial crystalline strata (Smith et al., 2019; Eisermann et al., 2020, in review). We can therefore hypothesize
that the spatial variations in subglacial strata also influence the position of present day grounding lines. Finally, the ramifications
of heterogeneous ocean bed properties go beyond ice volume considerations. Different levels of basal traction strongly affect
the magnitude of basal sliding. This in turn determines how much material is eroded underneath the ice sheet and transported
across the grounding line. As erosion rates are commonly approximated as basal sliding to some power (e.g. Herman et al.,
2015; Alley et al., 2019; Delaney and Adhikari, 2019), any differences in basal sliding velocities are exacerbated when erosion
volumes are computed. This uncertainty in eroded material produced has implications for how much sediment is available at
the ice bedrock interface and therefore if it is a hard or soft bed interface and its temporal variability.

## 5.3 Grounding line and ice sheet stability

The identified stable grounding line positions are not controlled by a single specific forcing alone, but are due to a combination
of sea level forcing, basal traction of the ocean bed, and ocean bathymetry. Other forcing mechanisms such as the SMB and





BMB are of secondary importance. However, the relative stability of grounding line position (<7 km of grounding line retreat) in the last 9,000 years of the retreat phase in both simulations coincides with the period of little sea level variations, leading us to conclude that at least for the retreat phase, sea level forcing is the most important model forcing. The modelled higher grounding line fidelity in the retreat phase for the soft bed can be attributed to the fact that ice discharge for the soft bed

simulation is dominated by basal sliding and higher ice velocities. In comparison, in the hard bed simulation ice discharge is dominated by internal deformation and almost no basal sliding, resulting in a much thicker ice sheet. This means that more ice needs to be removed before the grounded ice can detach from its subglacial material and initiate grounding line motion, thereby resulting in a much slower response time to changes in the model forcing. While our employed modelling approach makes it unlikely that the timing of our modelled stable grounding line positions are correct, they can still serve as rough spatial markers

of areas where depositional landforms such as grounding zone wedges or other geomorphological markers may be found.

### 5.4   Hysteresis of ice sheet simulations

The modelled grounding line advance in the first ∼300 years, we attribute to the fact that our ice sheet geometry is not completely in steady state after initialisation. This is due to inconsistencies of the model forcing (e.g. BMB parameterisation) in combination with boundary datasets (e.g. cavity topography). However, this does not affect our conclusions regarding ice

sheet hysteresis. Our results highlight the importance of different ocean bed properties onto the ice sheet's hysteresis behaviour. This underlines the dependence of the final ice sheet geometry on the model's initial state over timescales of a glacial cycle or longer. While bedrock geometry has long been identified as a cause for hysteresis behaviour in ice sheet models (e.g. Schoof, 2007) and remains an important indicator for future ice sheet vulnerability, our simulations show that in the absence of retrograde sloping bedrock topography, hysteresis can also be introduced by varying ocean bed properties. Despite very

similar model forcing, our simulations result in a non-linear response of ice sheet evolution that is exclusively controlled by ocean bed properties, revealing an additional challenging problem for model simulations over at least one advance and retreat cycle (Pollard and DeConto, 2009; Gasson et al., 2016). This also means that the employed modelling framework will likely not result in the correct ice sheet geometry at the LGM due to non-linear feedback mechanisms such as the marine ice sheet instability (Schoof, 2007; Durand et al., 2009), the height mass balance feedback (Oerlemans, 2002), and remaining

uncertainties regarding the subglacial topography.

### 5.5   Model limitations

The primary focus of the modelling framework was to extend the applicability of FS ice sheet models to glacial cycle timescales. This means that simplifications were made to other model components that we list here. We regard each of these simplification as a future avenue to improve upon the presented results.

The modelling approach presented here is tailored towards capturing ice and grounding line dynamics to high accuracy at the cost of comparatively naive parameterisations for the SMB and BMB which can be improved in the future. Also, by approximating hard and soft ocean beds through a time and space invariant friction coefficient, we omit spatial gradients in the thickness, grain size and cohesion of the ocean bed substrate. We therefore assume that properties of hard bed and soft bed





areas at the start of the simulation remain constant throughout the simulation. This means, areas in which little or enhanced basal sliding occurs in the modelling domain stay constant.

At the underside of the grounded ice sheet, we use a linear Weertman sliding law that relates the basal shear stress to the basal sliding velocity. In comparison to the non-linear Weertman sliding law, the linearised version has a tendency to reduce grounding line fidelity (e.g. Schannwell et al., 2018; Brondex et al., 2019). While this type of sliding law is still widely used (e.g. Ritz et al., 2015; Cornford et al., 2015; Nias et al., 2016; Yu et al., 2017; Schannwell et al., 2018; Brondex et al., 2019), pressure limited sliding relations (e.g. Tsai et al., 2015) are becoming more popular in the modelling community. The difference between Weertman and pressure limited relations is that the latter take effective pressure into account. This means that basal drag goes to zero near the grounding line and reduces to a plastic sliding relation (Brondex et al., 2017). However, this lower basal drag area is limited to a few kilometers upstream of the grounding line. Studies that have investigated the effect of the different sliding laws on grounding line retreat have found that the pressure limited relations lead to enhanced grounding line retreat (e.g. Schannwell et al., 2018; Brondex et al., 2019) in comparison to Weertman sliding laws. However, it is difficult to judge how much a pressure limited sliding law would affect our results as up to now no study has investigated this effect over an advance and retreat cycle.

Moreover, we have not considered glacial isostatic adjustment (GIA). Until recently, GIA was considered to be only important on timescales exceeding 1,000 years. However, recent progress has revealed that due to lower than previously assumed mantle viscosities, response times of GIA to ice unloading can be as short as five years for certain sections in Antarctica (Barletta et al., 2018; Whitehouse et al., 2019). While present day GIA rates for East Antarctica are relatively low (∼1mm/yr, see Martín-Español et al. (2016)) in comparison to regions of high mass loss in Antarctica, the effect over 20,000 years could amount to ∼20 m of elevation drop for the subglacial topography. This number is small in comparison to, for example, sea level variations (∼130 m), but may nevertheless result in a grounding line position that is not as far advanced at the glacial maximum as presented in our simulations.

## 6 Conclusions

Our simulations unlock a new time dimension for the applicability of FS ice sheet models on the regional scale. Application of an iterative solver reduced computation times in comparison to previous simulations by ∼80% and extended the temporal range of FS simulations by a factor of 40 compared to previous studies. This provides an important step towards including higher order physics into paleo ice sheet simulation and reduce uncertainties arising from approximations to the ice flow equations. Being able to simulate ice deformation to high accuracy over glacial timescales also opens opportunities for a better understanding of a number of subglacial processes (e.g. basal erosion).

We find ice volume differences of >50% over a glacial cycle that are exclusively caused by differing ocean bed properties. The different ocean bed properties also result in different ice sheet growth and decay pattern with the thick and slow flowing hard bed simulation exhibiting strong hysteresis behaviour. This is completely absent in the thin and fast flowing soft bed simulation. As recent geophysical observations (e.g. Gohl et al., 2013; Smith et al., 2019; Eisermann et al., 2020, in review)



indicate a more hetereogenous substrate distribution (sediments vs. crystalline bedrock) than previously thought, this could have important consequences for past stable ice sheet geometries and grounding line positions as well as for the present and future response of the ice sheet's grounding line to ocean warming.

*Code availability.* The Elmer/Ice code is publicly available through GitHub (https://github.com/ElmerCSC/elmerfem, lastaccess: 05 November 2019). All simulations were performed with version 8.3 (rev. 74a4936). Elmer/Ice scripts including all necessary input files to reproduce the simulations are available at https://doi.org/10.5281/zenodo.3564168 (Schannwell (2019), last access: 28 February 2020).

*Video supplement.* There is one video supplement SI video 1

*Author contributions.* CS and RD conceived the study with input from OE, TAE, and CM. Simulations were run by CS. E.C.S and H.E. provided new cavity topography data. The manuscript was written by CS and RD, and all authors contributed to editing and revision.

*Competing interests.* Olaf Eisen is a co-editor in chief of TC

*Acknowledgements.* Clemens Schannwell was supported by the Deutsche Forschungsgemeinschaft (DFG) grant EH329/11-1 (to TAE) in the framework of the priority programme "Antarctic Research with comparative investigations in Arctic ice areas". Reinhard Drews was funded in the same project under MA 3347/10-1. Reinhard Drews is supported by the DFG Emmy Noether grant DR 822/3-1. The authors gratefully acknowledge the Gauss Centre for Supercomputing e.V. (www.gauss-centre.eu) for providing computing time on the GCS Supercomputer SuperMUC-NG at Leibniz Super- computing Centre (www.lrz.de).



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
