# Peer review of "Quantifying the effect of ocean bed properties on ice sheet geometry over 40,000 years with a full-Stokes model"

_The Cryosphere, 2020_

## Referee Comment (RC1) · Anonymous Referee #1 · 23 May 2020

In this paper, the authors use a state-of-the-art ice sheet model to carry out (slightly compressed) glacial interglacial simulations on the Ekstrom ice catchment and ice shelf. They use a stokes ice-sheet model with a grounded to floating transition and step forward the ice-sheet model over time scales considerably longer than those generally considered in such simulations. By make use of two different linear system solvers, as well as testing the response to a highly uncertain physical parameter dealing with the nature of the sea bottom in the ice-shelf cavity (at least, this is my understanding, see below). The improved numerical solver shows great gains in terms of computational cost, and the bed geology of ice-shelf cavities is shown to play a large role in fluctuation of ice-sheet volume and potentially lead to hysteretic behaviour.

I believe this paper can potentially by published in this journal. The simulations they have done are impressive from a computational standpoint alone, but also raise interesting questions regarding our ability to model past behaviour of ice sheets when we know so little about the marine geology of ice-shelf cavities. I have a few general comments, and a number of detailed comments, however, that should ideally be addressed.

General Comments:

1. Though this is quite specific, it is quite important, and i would like to see it clarified, ideally in the response and in a revision. The central science result hinges around the effect of different bed strength. The methods seems to suggest that, between the "hard bed" and "soft bed" runs, the only difference between the bed frictional coefficient (C) is in areas where this CANNOT be inferred from an inversion of velocities as described in 3.4 — in other words, bed within the current ice-shelf cavity — meaning in both experiments, C is identical in currently grounded areas. Is this correct??? I ask because section 3.4 would imply this, though i could not find any other part of the paper that made this clear. If the only difference is indeed below currently floating ice, this has very strong implications; however, i fear that (a) i have misunderstood and (b) even if i have not, other readers might. This aspect of the methodology should be pointed out with crystalline clarity to the point that maybe even the experiment names should change to emphasise this.

(And i should add if "hard bed" means 10-1 everywhere and soft bed means 10-5 everywhere, then the results overall are not very surprising — so this is why it is general point #1)

2. A key scientific result put forth is that of hysteresis with a strong (ice shelf cavity?) bed, in that the grounding line (GL) does not return to its original position. This is used to argue that even without a retrograde slope (line 364) there can be hysteretic behaviour. I point out that there have been previous studies suggesting that a continuum of grounding lines were possible, but these (and other) authors later showed that correct treatment of grounding zone boundary layers removed this degeneracy, but this treatment involved resolving the grounding zone, the length of which scales inversely with bed strength. In the context of Stokes, Nowicki and Wingham (2008) found that with an effectively non-sliding bed, there was not a unique sliding solution in the presence of a frozen grounding zone. While the authors' results are interesting, they should allow for the possibility that (a) the grounding zone is not sufficiently resolved or (b) there is not a unique solution to the Stokes contact problem with an effectively non sliding bed (therefore raising the question of whether the model finds the physically correct one) rather than assuming that the model results are correct, and hysteresis of ice sheet is possible without retrograde beds.

3. There are extensive mentions of ensemble modelling in the paper; while you do not say outright you are doing ensemble modelling, you don't say that you are not (aside from a mention that your approach is "complimentary" to it, line 310, which is confusing; it is not ensemble modelling because it does not vertically average?) I would argue you tested 2 end members of a (albeit important) physical parameter (the choice of solver is not a physical parameter), so perhaps you should be as clear as you can be that this is NOT an ensemble of experiments

4. A series of 4 experiments are done, varying one of each: bed strength, and numerical solver — and the results of the paper are presented as dual: the effects of the hard bed, and the effectiveness of the solver. It is therefore confusing whether this paper is meant to be about numerics (in which case it might be better suited for a different journal) or about the scientific results. Both aspects are presented quite prominently making the message of the paper a bit unclear. If the paper is to be about science, then aspects dealing with numerical methods such as scaling should perhaps be in an appendix and not feature in the abstract (though i do have comments about these aspects as well).

Detailed comments:

[Figure]

line 73, data of shelf cavity — should point out this is only relevant to the present study *up to the farthest point of grounding line advance* in your experiments.

line 174 and potentially elsewhere; please say something about the FEM basis functions in your scheme(s). It is important to establish that the basis functions are LBB conforming and that the solutions are exactly mass conserving (ie. not using penalty methods) — the latter perhaps not being as important for short term runs but very important for long term.

line 174 how many cells? how many DoFs?

line 203: are you sure? all physical uncertainty? what about ice shelf crevassing weakening? Not to mention these physical parameters, if i understand correctly, are only varied in the ice-shelf cavity (see General Comment 1)

line 223: since this is exceptional is it really of value for general knowledge? also:

a) it is odd to compare one solver on one system and another on another system. how about an additional test (only a few time steps) of both on the same system, with wall times so a comparison can be made. b) ParStokes has great scaling but what about absolute time for a fixed core count on the same system compared against MUMPS?

line 228: following on from comment on line 174, which of the two uses a stabilisation method? if not both, then what about the other one?

Figure 8: Here or in an appendix you should show a similar plot comparing ParStokes and Mumps for soft and hard bed (whichever shows poorer agreement). It is important to establish that the effect of solver, while having a large difference on performance, has very small effects on Volume and GL position relative to the effect of the physical parameter. If MUMPS and ParStokes differ, at most one is correct — if the difference is large, how are we to trust the physical results?

line 250: following what?

line 257: then there is a volume decrease — what causes it?

line 289, some funny maths. How does increasing performance by a factor of 6 allow runs of 40,000yrs when using the MUMPS solver allowed less than 1000a? this is not a factor of 6. did you do something to increase the time step that was not mentioned?

line 349: explain what you mean by grounding line "fidelity"

Refs:

Nowicki, S. M. J., and D. J. Wingham (2008), Conditions for a steady ice sheet-ice shelf junction, Earth Planet. Sci. Lett., 265, 246–255

––––––––––––––––––––––––––––––

---

## Referee Comment (RC2) · Stephen Cornford (Referee) · 29 Jul 2020

This paper describes the application of a full-Stokes ice sheet model to a modest sized region (about 100 x 300 km) of Antarctica over 40,000 years at a reasonable 1km resolution. Full Stokes models are computationally expensive and have typically been used only for shorter simulations: various approximations are normally applied (quite often at coarser resolution too). The paper also makes use of new data and explores the importance of bed friction in the region, so would be of interest even if it did not manage the full Stokes model. Give the use of the higher-fidelity model, this is an important and clearly written paper. I have a few minor comments only.

[Figure]

Comments

The abstract perhaps emphasises the Stokes model, but there are two conclusions in the paper – one relates to the importance of the basal boundary condition (sliding law), which might have been reached with a more approximate model. At the same time, there is no less-than-Stokes model considered, so the paper provides us with no information on whether 'uncertainties due to physical approximations [have been] be reduced.', at least compared to the uncertainties that would be common to models (e.g the sliding law)

Specific comments

L42: "The rationale behind this tuning is that if the model matches the constraints well, then confidence is high that the model also reproduces ice sheet changes at other times. The risk involved is that the matching may overcompensate for the simplified model physics leading to higher uncertainties in future predictions where model constraints are absent" I don't disagree with the overall statement, but I would suggest that the rationale is simply that if a model matches constraints poorly, then it should be rejected (or given a lower score).

L105; The thermodynamic equation – how is temperate ice treated?

L153 "A linear viscous sliding relation (m=1) was chosen to guarantee consistency between model intialisation and forcing simulation." This is not needed – the inverse problem provides both $C_1$ and $|u_b|$, so you could carry out runs. with any value of $m$ so long as $C_1|u_b| = C_m|u_b|^m$. Linear sliding is probably the worst choice (see e,g Joughin 2010) and although many (me included) have used these rules in the past, as a community we should move on. I am not suggesting new runs, but an acknowledgement that the authors understand this position.

L183 "While robust, direct solvers do not take advantage of the sparse structure of the matrix and require large amounts of memory." That is certainly true of e.g. LA-

[Figure]

PACK solvers, but the MUMPS solver is the MUltifrontal Massively Parallel sparse direct Solver, designed for these sorts of problems. That is not to say that an iterative solver has no advantages, but frontal solvers like MUMPS are specialised over general dense solvers.

L226 "We note however that we do not expect a perfect match between the two solver setups due to small differences in the finite element formulation" This needs a bit more emphasis/elaboration. If you were solving the same problem, you would expect the solvers to give the same results (assuming the iterative technique was successful). But the problems are different? ParStokes does seem to work well though (I would have liked see SSA in the same comparison, but in a follow up paper, perhaps)

L220 "For both simulations, there is good agreement in terms of grounding line position over time, with differences never exceeding 5% (Figure 5)." - the difference is in total grounded area.

L264: "Stable grounding line positions for both simulations are associated with periods of ice sheet stability (Fig. 8). " Steady rather than stable? I agree that you are unlikely to see unstable equilibrium in practice, so steadiness tends to imply stability.

L289 "The high mesh resolution required to adequately capture grounding line migration (Pattyn et al., 2013) is hereby maintained.". Perhaps – there is no convergence study in this paper so it relies on external references, and the only Stokes model in Pattyn 2013 is Elmer/Ice which ran at around 50m resolution.

L385 "The difference between Weertman and pressure limited relations is that the latter take effective pressure into account. This means that basal drag goes to zero near the grounding line and reduces to a plastic sliding relation (Brondex et al., 2017). However, this lower basal drag area is limited to a few kilometers upstream of the grounding line. " There is another important difference, which is the independence of $T_b$ and $|u|$ in the region in question, which could be substantial. See for example Joughin 2019 which provides evidence for Coulomb-like sliding a long way from

the grounding line. No need to speculate, but please, acknowledge Joughin 2019 Âă https://doi.org/10.1029/2019GL082526

---

## Author Comment (AC1) · 2 Sep 2020

**Reviewer 1:**

In this paper, the authors use a state-of-the-art ice sheet model to carry out (slightly compressed) glacial interglacial simulations on the Ekstrom ice catchment and ice shelf. They use a stokes ice-sheet model with a grounded to floating transition and step forward the ice-sheet model over time scales considerably longer than those generally considered in such simulations. By make use of two different linear system solvers, as well as testing the response to a highly uncertain physical parameter dealing with the nature of the sea bottom in the ice-shelf cavity (at least, this is my understanding, see below). The improved numerical solver shows great gains in terms of computational cost, and the bed geology of ice-shelf cavities is shown to play a large role in fluctuation of ice-sheet volume and potentially lead to hysteretic behaviour.

I believe this paper can potentially by published in this journal. The simulations they have done are impressive from a computational standpoint alone, but also raise interesting questions regarding our ability to model past behaviour of ice sheets when we know so little about the marine geology of ice-shelf cavities. I have a few general comments, and a number of detailed comments, however, that should ideally be addressed.

We thank the referee for the thoughtful and thorough review of our paper. We appreciate you taking the time to complete the reviews and welcome your helpful comments. We have revised the manuscript to address your review comments (see below). Throughout this response to review document your (referee review) comments are provided in regular, non-italic font text, our response comments are provided in red font (as here).

General Comments:

1. Though this is quite specific, it is quite important, and i would like to see it clarified, ideally in the response and in a revision. The central science result hinges around the effect of different bed strength. The methods seems to suggest that, between the "hard bed" and "soft bed" runs, the only difference between the bed frictional coefficient (C) is in areas where this CANNOT be inferred from an inversion of velocities as described in 3.4 ă ˘A˘T in other words, bed within the current ice-shelf cavity ă ˘A˘T meaning in both experiments, C is identical in currently grounded areas. Is this correct???

Yes, this is correct. We are only investigating the effect of different geology underneath present-day ice shelves (floating parts), while the inferred distribution of the friction coefficient (C) for the present-day ice sheet (grounded parts) is the same in all simulations.

I ask because section 3.4 would imply this, though i could not find any other part of the paper that made this clear. If the only difference is indeed below currently floating ice, this has very strong implications; however, i fear that (a) i have misunderstood and (b) even if i have not, other readers might. This aspect of the methodology should be pointed out with crystalline clarity to the point that maybe even the experiment names should change to emphasise this. (And i should add if "hard bed" means 10-1 everywhere and soft bed means 10-5everywhere, then the results overall are not very surprising â˘A˘T so this is why it is general point #1)

To make this more clear, we have added the following to the introduction: "Here, we present the first regional scale FS simulations investigating the effect of different ocean bed properties under contemporary ice shelves on ice sheet geometry over a glacial cycle."
We also added the following statement to the model initialisation: "The model is initialised to the present day geometry using the commonly applied snapshot initialisation in which the basal traction coefficient C is inferred under the grounded ice sheet by matching observed surface velocities with modelled surface velocities."

Please note that in the Experimental Design section, we also explicitly state that different values for basal traction are only applied underneath present day ice shelves: "We consider two end member basal property scenarios by prescribing either soft ocean bed conditions (mimicking sediment deposits) or hard ocean bed conditions (mimicking crystalline rock) under all present day ice shelves in the modelling domain."

2. A key scientific result put forth is that of hysteresis with a strong (ice shelf cavity?) bed, in that the grounding line (GL) does not return to its original position. This is used to argue that even without a retrograde slope (line 364) there can be hysteretic behaviour. I point out that there have been previous studies suggesting that a continuum of grounding lines were possible, but these (and other) authors later showed that correct treatment of grounding zone boundary layers removed this degeneracy, but this treatment involved resolving the grounding zone, the length of which scales inversely with bed strength. In the context of Stokes, Nowicki and Wingham (2008) found that with an effectively non-sliding bed, there was not a unique sliding solution in the presence of a frozen grounding zone. While the authors' results are interesting, they should allow for the possibility that (a) the grounding zone is not sufficiently resolved or (b) there is not a unique solution to the Stokes contact problem with an effectively non sliding bed (therefore raising the question of whether the model finds the physically correct one) rather than assuming that the model results are correct, and hysteresis of ice sheet is possible without retrograde beds.

We thank the reviewer for pointing this out. We have added the following sentence to the discussion section of the modelled hysteresis to acknowledge this: "However, this result could also be caused by a combination of the non-uniqueness of the Stokes contact problem for non-sliding beds and an under resolving of the grounding line zone (e.g. Nowicki and Wingham, 2008)."

3. There are extensive mentions of ensemble modelling in the paper; while you do not say outright you are doing ensemble modelling, you don't say that you are not (aside from a mention that your approach is "complimentary" to it, line 310, which is confusing; it is not ensemble modelling because it does not vertically average?) I would argue you tested 2 end members of a (albeit important) physical parameter (the choice of solver is not a physical parameter), so perhaps you should be as clear as you can be that this is NOT an ensemble of experiments.

The reviewer is correct that we are not presenting ensemble simulations, but provide a pair of envelope simulations covering two extreme scenarios. We have added a sentence to the introduction to state this explicitly. It reads: "We do this by investigating end-member scenarios as opposed to ensemble modelling."

We also expanded on the statement in line 310 now explicitly stating that our approach of using a complex ice-physics model investigating end-member scenarios and ensemble modelling using simplified ice physics both have their advantages and disadvantages, but both are worth pursuing.

It now reads: "For example, radar isochrones for floating ice shelves could be incorporated more easily into the model tuning, because the FS approach does not apply a vertical average in these areas unlike ice models using a simplified force balance. We believe that ensemble modelling using simpler ice physics models and our approach of employing a complex ice-physics model and investigating end-member scenarios can both provide different new insights. Hence, both approaches should be pursued in future. This also holds for shallow ice approximation-FS hybrid approaches (Ahlkrona et al., 2016) which can build on the results shown here."

A series of 4 experiments are done, varying one of each: bed strength, and numerical solver ậ˘AˇT and the results of the paper are presented as dual: the effects of the hard bed, and the effectiveness of the solver. It is therefore confusing whether this paper is meant to be about numerics (in which case it might be better suited for a different journal) or about the scientific results. Both aspects are presented quite prominently making the message of the paper a bit unclear. If the paper is to be about science, then aspects dealing with numerical methods such as scaling should perhaps be in an appendix and not feature in the abstract (though i do have comments about these aspects as well).

The goal of the paper is indeed two-fold. One goal is to show that progress towards faster full-Stokes simulations is being made, so that in the near future simulations over longer time scales (>1,000 years) with this type of model should be possible for regional domains. As the scaling of the ParStokes solver is integral to the speed-up in computation time, we would like to keep this Figure in the main text.
The second goal of the paper is to present the effect of different geology under present-day ice shelves for ice-sheet evolution over a glacial cycle. The main scientific messages are however independent of the solver setup. We tried to make

this separation clear by putting technical and scientific results and their discussion in separate sections. However, to make our intentions clear to the reader from the beginning, we have rewritten the last paragraph of the introduction.

It now reads: "Here, we present the first regional scale FS simulations investigating the effect of different ocean bed properties under contemporary ice shelves on ice sheet geometry over a glacial cycle. We do this by investigating end-member scenarios as opposed to ensemble modelling. This means, we specify either very soft and slippery or very hard and sticky conditions under present-day ice shelves. The goal of the paper is hence two-fold. First, we present methodological advances by extending the feasibility of regional FS ice sheet simulations by an order of magnitude using the open source code Elmer/Ice (Gagliardini et al., 2013). We do this with a highly parallelised numerical scheme allowing to maintain a high mesh resolution (~1 km) and a freely evolving grounding line over glacial/interglacial timescales. Second, we present new scientific insights regarding the effect of different ocean bed properties seawards of today's grounding line and quantify its impact on the evolution of the entire catchment. This is done for the Ekström Ice Shelf catchment, Dronning Maud Land, East Antarctica (Fig. 1)."

Detailed comments:

line 73, data of shelf cavity â˘A˘T should point out this is only relevant to the present study *up to the farthest point of grounding line advance* in your experiments.

Correct. We added the following sentence: "For our simulations, this difference is only relevant up to the point of farthest grounding line advance."

line 174 and potentially elsewhere; please say something about the FEM basis functions in your scheme(s). It is important to establish that the basis functions are LBB conforming and that the solutions are exactly mass conserving (ie. not using penalty methods) â˘A˘T the latter perhaps not being as important for short term runs but very important for long term.
line 174 how many cells? how many DoFs?

We have added this information to the mesh generation and refinement section.

It now reads: "The 3D mesh consists of ~200,000 nodes and therefore ~800,000 degrees of freedom. We are using stabilised P1P1 elements and an algorithm that deduces a mass-conserving nodal surface to avoid artificial mass loss (Gagliardini et al., 2013)."

line 203: are you sure? all physical uncertainty? what about ice shelf crevassing weakening? Not to mention these physical parameters, if i understand correctly, are only varied in the ice-shelf cavity (see General Comment 1)

The reviewer is correct. We qualified this statement. It now reads: " Hence they also account for some uncertainties in model parameters, forcings, and physics of the applied ice sheet model."

line 223: since this is exceptional is it really of value for general knowledge?

We agree that there is not much value for the compute times using a now decommissioned system. We therefore removed the paragraph from the manuscript and also deleted the corresponding compute times in Figure 4.

Also:

a)  it is odd to compare one solver on one system and another on another system. How about an additional test (only a few time steps) of both on the same system, with walltimes so a comparison can be made.

We agree that this would be the preferred option. However, larger jobs get priority on the SuperMUC-NG and hence queuing times would be much longer on this system for smaller jobs like the MUMPS jobs. We did however test whether absolute compute times for a few time steps between the systems are similar. As this was the case, we believe that the absolute numbers provided in Figure 4 are most informative.

b) ParStokes has great scaling but what about absolute time for a fixed core count on the same system compared against MUMPS?

We added the following statements to the solver setup comparison section (sec 4.1): "This speed-up is in part due to using more CPUs in the ParStokes simulations. When comparing absolute runtime of the scaling simulations, ParStokes provides faster computations for >168 CPUs. This means the minimum requirement for faster simulations with ParStokes is a supercomputer with more than 168 CPUs. The exact CPU number may however very well vary from system to system depending on the available hardware."

line 228: following on from comment on line 174, which of the two uses a stabilisation method? if not both, then what about the other one?

They both use stabilisation methods, but  different ones (stabilized method for MUMPS and bubble stabilisation for ParStokes) for stability reasons. We added the following sentence: " When using the MUMPS solver, the stabilised method is used, while for the ParStokes solver we use bubble stabilisation (Gagliardini et al., 2013). This results in slightly different systems that need to be solved."

Figure 8: Here or in an appendix you should show a similar plot comparing ParStokes and Mumps for soft and hard bed (whichever shows poorer agreement). It is important to establish that the effect of solver, while having a large difference on performance, has very small effects on Volume and GL position relative to the

effect of the physical parameter. If MUMPS and ParStokes differ, at most one is correct ˘A˘T if the difference is large, how are we to trust the physical results?

We do not think that this proposed Figure would add information that is not already there. We are showing grounded area differences in Figure 5 which is indirectly a measure for the differences in ice volume. In addition, Figures 6 and 7 show differences in grounding-line position and ice thickness between the solver setups for different time snapshots. We believe that these Figures provide a solid case that differences between the solver setups are small.

line 250: following what?

Changed to: "Following the period of volume gain, ..."

line 257: then there is a volume decrease ˘A˘T what causes it?

Apologies, but we could not find the statement of a volume decrease the reviewer is referring to.

line 289, some funny maths. How does increasing performance by a factor of 6 allow runs of 40,000yrs when using the MUMPS solver allowed less than 1000a? this is not a factor of 6. did you do something to increase the time step that was not mentioned?

I think we confused the reviewer here. These numbers have nothing to do with each other. The speed-up refers to the difference in computation time between MUMPS and ParStokes. MUMPS does allow computations of 40,000 years, they just take 3-6 times as long as with ParStokes. But given enough time, these type of runs are possible. To avoid this confusion we removed the 1,000 year maximum from the sentence.
It now reads: "The new setup allows 3D full-Stokes ice sheet simulations
on the regional scale over 40,000 years now in under a month's time."

line 349: explain what you mean by grounding line
We changes this to: "The earlier onset of grounding line motion in the retreat phase ..."

---

## Author Comment (AC2) · 2 Sep 2020

**Reviewer 2 (Stephen Cornford):**

This paper describes the application of a full-Stokes ice sheet model to a modest sized region (about 100 x 300 km) of Antarctica over 40,000 years at a reasonable 1km resolution. Full Stokes models are computationally expensive and have typically been used only for shorter simulations: various approximations are normally applied (quite often at coarser resolution too). The paper also makes use of new data and explores the importance of bed friction in the region, so would be of interest even if it did not manage the full Stokes model. Give the use of the higher-fidelity model, this is an important and clearly written paper. I have a few minor comments only.

We thank the referee for the thoughtful and thorough review of our paper. We appreciate you taking the time to complete the reviews and welcome your helpful comments. We have revised the manuscript to address your review comments (see below). Throughout this response to review document your (referee review) comments are provided in regular, non-italic font text, our response comments are provided in red font (as here).

The abstract perhaps emphasises the Stokes model, but there are two conclusions in the paper – one relates to the importance of the basal boundary condition (sliding law), which might have been reached with a more approximate model. At the same time, there is no less-than-Stokes model considered, so the paper provides us with no information on whether 'uncertainties due to physical approximations [have been] be reduced.', at least compared to the uncertainties that would be common to models (e.g the sliding law)

We agree that our paper does not show that uncertainties due to different physical approximations have been reduced. Rather, the goal of the paper is to provide a first step towards being able to do this in the near future. To reflect this appropriately in the text, we changed the corresponding sentence to: "Therefore, there is a need to extend the applicability of regional FS ice sheet models to timescales longer than 1,000 years so that uncertainties due to physical approximations in the force balance can be quantified and reduced in the near future."
We also agree that at least qualitatively, we could have reached the same conclusions regarding different levels of bed friction with a more approximate model. However, the magnitude of grounding-line advance and retreat over such a long time period will most likely be different across different ice mechanical models. This has been shown in the previous intercomparison studies using idealised geometries (e.g. Pattyn et al. 2013).

Specific comments:

L42: "The rationale behind this tuning is that if the model matches the constraints well,then confidence is high that the model also reproduces ice sheet changes at other times. The risk involved is that the matching may overcompensate for the

simplified model physics leading to higher uncertainties in future predictions where model constraints are absent" I don't disagree with the overall statement, but I would suggest that the rationale is simply that if a model matches constraints poorly, then it should be rejected (or given a lower score).

Agreed. We changed this accordingly.

L105; The thermodynamic equation – how is temperate ice treated?

We added the following: "The ice temperature $T$ is bounded by the pressure melting point $T_m$, so that $T \leq T_m$."

L153 "A linear viscous sliding relation (m=1) was chosen to guarantee consistency between model intialisation and forcing simulation." This is not needed – the inverse problem provides both C1 and |ub|, so you could carry out runs. with any value of m so long as C1|ub|=Cm|ub|m. Linear sliding is probably the worst choice (see e.g Joughin 2010) and although many (me included) have used these rules in the past, as a community we should move on. I am not suggesting new runs, but an acknowledgement that the authors understand this position.

Yes, we are aware of this and agree with the reviewer here. We changed this sentence to read: "A linear viscous sliding relation (m=1) was chosen. Alternative and physically more realistic sliding relations exits (e.g. Joughin et al., 2019) and the consequences of our choice of using a linear sliding relation on the results are discussed below (see section 5.5)."

L183 "While robust, direct solvers do not take advantage of the sparse structure of the matrix and require large amounts of memory." That is certainly true of e.g. LAPACK solvers, but the MUMPS solver is the MUltifrontal Massively Parallel sparse direct Solver, designed for these sorts of problems. That is not to say that an iterative solver has no advantages, but frontal solvers like MUMPS are specialised over general dense solvers.

We agree with the reviewer here. Our formulation was not precise enough. MUMPS is certainly tailored towards solving large sparse linear systems. However, the fact that it remains a direct solver still leads to the solver being memory bounded. Therefore, it does not scale at all beyond 80 CPUs. We adjusted the sentence as follows: "While robust, direct solvers require large amounts of memory."

L226 "We note however that we do not expect a perfect match between the two solver setups due to small differences in the finite element formulation" This needs a bit more emphasis/elaboration. If you were solving the same problem, you would expect the solvers to give the same results (assuming the iterative technique was successful). But the problems are different?

Yes, the problems are slightly different due to different stabilisation methods employed by using MUMPS or ParStokes (see response to other reviewer).

ParStokes does seem to work well though (I would have liked see SSA in the same comparison, but in a follow up paper, perhaps).

We agree that this would have been interesting. However, as of today there is no thermomechanical coupling available when using reduced models in Elmer/Ice (e.g. SSA, SSA*) and that's why we did not perform the same simulations with a reduced model.

L220 "For both simulations, there is good agreement in terms of grounding line position over time, with differences never exceeding 5% (Figure 5)." - the difference is in total grounded area.

Yes. Thanks for spotting this. Changed accordingly.

L264: "Stable grounding line positions for both simulations are associated with periods of ice sheet stability (Fig. 8). " Steady rather than stable? I agree that you are unlikely to see unstable equilibrium in practice, so steadiness tends to imply stability.

Yes, steady might be the better term to use here. Changed accordingly.

L289 "The high mesh resolution required to adequately capture grounding line migration (Pattyn et al., 2013) is hereby maintained.".
Perhaps – there is no convergence study in this paper so it relies on external references, and the only Stokes model in Pattyn 2013 is Elmer/Ice which ran at around 50 m resolution.

The reviewer is correct that we did not perform a convergence study. Given the runtime of the model, we do not think it is feasible to carry out a convergence study for long-term simulations at the moment. Moreover, a mesh resolution of 50 m is certainly only ever applied in simplified settings and for shorter simulation times. To acknowledge the fact that we cannot show that this resolution is adequate, we reformulated the sentence as follows: "We hereby maintain a mesh resolution (~1 km) that is finer than in most other paleo ice sheet simulations (Pollard and DeConto, 2009; Golledge et al., 2014; Albrecht et al., 2020) albeit at a regional scale."

L385 "The difference between Weertman and pressure limited relations is that the latter take effective pressure into account. This means that basal drag goes to zero near the grounding line and reduces to a plastic sliding relation (Brondex et al., 2017). However, this lower basal drag area is limited to a few kilometers upstream of the grounding line."
There is another important difference, which is the independence of Tb and |u| in the region in question, which could be substantial. See for example Joughin

2019 which provides evidence for Coulomb-like sliding a long way from the grounding line. No need to speculate, but please, acknowledge Joughin 2019 Ăahttps://doi.org/10.1029/2019GL082526

We have expanded this section and added the reference. It now reads: "The difference between Weertman and pressure limited relations is that the latter take effective pressure into account. This means that basal drag goes to zero near the grounding line and reduces to a plastic sliding relation (Brondex et al., 2017). This results in the basal drag becoming independent of the sliding velocity. Most previous studies using pressure-limited relations confine areas of lower basal drag to within a few kilometers upstream of the grounding line (e.g. Schannwell et al., 2018; Brondex et al., 2019). There is however evidence from observations and modelling that areas of low basal drag can extend much farther inland (Joughin et al., 2019)."

Pattyn, F., Perichon, L., Durand, G., Favier, L., Gagliardini, O., Hindmarsh, R., . . . Wilkens, N. (2013). Grounding-line migration in plan-view marine ice-sheet models: Results of the ice2sea MISMIP3d intercomparison. Journal of Glaciology, 59(215), 410-422. doi:10.3189/2013JoG12J129